# Chitosan Scaffolds from Crustacean and Fungal Sources: A Comparative Study for Bone-Tissue-Engineering Applications

**DOI:** 10.3390/bioengineering11070720

**Published:** 2024-07-16

**Authors:** Neelam Iqbal, Payal Ganguly, Lemiha Yildizbakan, El Mostafa Raif, Elena Jones, Peter V. Giannoudis, Animesh Jha

**Affiliations:** 1School of Chemical and Process Engineering, University of Leeds, Leeds LS2 9JT, UK; 2Faculty of Medicine and Health, School of Dentistry, University of Leeds, Leeds LS2 9JT, UK; e.m.raif@leeds.ac.uk; 3Leeds Institute of Rheumatic and Musculoskeletal Medicine, University of Leeds, Leeds LS9 7JT, UK; 4Academic Department of Trauma and Orthopaedic Surgery, University of Leeds, Leeds LS2 9JT, UK

**Keywords:** fungal, crustacean, chitosan, bone regeneration, tissue engineering

## Abstract

Chitosan (CS), a biopolymer, holds significant potential in bone regeneration due to its biocompatibility and biodegradability attributes. While crustacean-derived CS is conventionally used in research, there is growing interest in fungal-derived CS for its equally potent properties in bone regenerative applications. Here, we investigated the physicochemical and biological characteristics of fungal (MDC) and crustacean (ADC)-derived CS scaffolds embedded with different concentrations of tricalcium phosphate minerals (TCP), i.e., 0(wt)%: ADC/MDC-1, 10(wt)%: ADC/MDC-2, 20(wt)%: ADC/MDC-3 and 30(wt)%: ADC/MDC-4. ADC-1 and MDC-1 lyophilised scaffolds lacking TCP minerals presented the highest zeta potentials of 47.3 ± 1.2 mV and 55.1 ± 1.6 mV, respectively. Scanning electron microscopy revealed prominent distinctions whereby MDC scaffolds exhibited striation-like structural microarchitecture in contrast to the porous morphology exhibited by ADC scaffold types. With regard to the 4-week scaffold mass reductions, MDC-1, MDC-2, MDC-3, and MDC-4 indicated declines of 55.98 ± 4.2%, 40.16 ± 3.6%, 27.05 ± 4.7%, and 19.16 ± 5.3%, respectively. Conversely, ADC-1, ADC-2, ADC-3, and ADC-4 presented mass reductions of 35.78 ± 5.1%, 25.19 ± 4.2%, 20.23 ± 6.3%, and 13.68 ± 5.4%, respectively. The biological performance of the scaffolds was assessed through in vitro bone marrow mesenchymal stromal cell (BMMSCs) attachment via indirect and direct cytotoxicity studies, where all scaffold types presented no cytotoxic behaviours. MDC scaffolds indicated results comparable to ADC, where both CS types exhibited similar physiochemical properties. Our data suggest that MDC scaffolds could be a potent alternative to ADC-derived scaffolds for bone regeneration applications, particularly for 10(wt)% TCP concentrations.

## 1. Introduction

Bone-tissue-engineering (BTE) has emerged as a vital subfield of tissue engineering, aiming to develop effective therapeutic strategies for bone defects resulting from trauma, congenital abnormalities, or diseases such as osteoporosis and bone cancer. The current standard treatment for bone defects involves autografts, allografts, and synthetic materials. However, these approaches are associated with limitations, including donor site morbidity, risk of infection, and inadequate graft integration. As a result, there is growing interest in identifying and characterising novel biomaterials, such as chitosan (CS), for BTE applications. CS’s unique physicochemical and biological properties provide an attractive candidate for BTE applications. CS is a copolymer obtained from a thermochemical partial deacetylation process of the polysaccharide chitin (CT) [1,2,3]. Deacetylation involves the chemical hydrolysis of the acetyl groups under alkaline conditions (concentrated sodium hydroxide (NaOH)) [4,5] or by enzymatic hydrolysis via CT deacetylase [4,6]. The resulting structure consists of hydrophilic primary amino groups conferring positive charges [7]. 

The source (crustaceans, squid, fungi, etc.) [7] and processing conditions of CT affect the molecular weight (M_w_), for example, the number of amino (-NH_2_) and hydroxyl (OH) functional groups formed [7,8]. Despite having an identical fundamental chemical structure, CS derived from crustacean and fungal sources presents numerous substantial variations that may impact their respective functionalities, properties, and applications. The most traditional source of CS is crustaceans, i.e., from seafood processing, including crab, lobster, and shrimp shells, whereby the diversity of crustacean sources often results in high batch variability, which affects the degree of deacetylation (DDA) and the molecular weight (M_w_). The chemical manufacturing of CS requires harsh and time-consuming processes and leads to the formation of additional unwanted products such as calcium carbonates, pigments, and proteins. Thus, the amount of CS produced is often reduced [3,9], and the lack of consistent reproducibility ultimately affects the physiochemical and biological properties of CS derived from crustacean sources. Crustacean-derived CS (ADC) is also known to elicit allergic reactions in individuals with shellfish allergies [10], increasing the need for alternative CS sources.

Conversely, other origins of CS include fungal sources, i.e., zygomycetes and insects. Fungi are cultivated under controlled conditions; thus, the fungi-derived CS is generally of higher purity with consistent physicochemical properties [11]. More importantly, CS from fungal sources demonstrates a lower percentage of minerals and impurities correlating to reduced allergic contaminants, offering significantly lower health risks than CS obtained from crustacean sources [12]. Fungal cell walls contain CT and CS, whereby the CS extraction process only involves a weak acid treatment, thus having a limited or no adverse effect on the polysaccharide and reducing waste generation. Fungal sources for chitosan (e.g., Aspergillus Niger) are abundant and sustainable, which is not always the case with crustacean sources (e.g., crustaceans). CS derived from fungi can be produced year-round in controlled conditions, unlike crustacean-derived chitosan, which can depend on the seafood industry and seasonal fluctuations. Fungal sources of CS are increasingly being used for potential applications such as neutraceuticals, biocoagulent for wastewater treatment [13], within the food industry [14] and for wound-healing purposes [15]. Fungal-derived CS has been shown to possess properties similar to crustacean-derived CS but with a lower antigen effect, the ability to dissolve at physiological pH ranges, and the ability to be used as a non-viral gene delivery system [9]. Although fungal-derived CS has ecological and non-allergic advantages, the limiting factor seems related to the extraction and processing techniques, resulting in lower yields than crustacean-derived CS [16]. 

Recent studies have highlighted the role of CS’s origin in modulating its properties relevant to BTE, such as porosity, mechanical strength, and bioactivity [17]. Both fungal- and crustacean-derived CS have demonstrated the potential to promote bone regeneration through their osteoconductive and osteoinductive capabilities [18]. For instance, CS scaffolds derived from different sources can exhibit porosity and pore size variations, directly impacting cellular infiltration, nutrient diffusion, and bone tissue ingrowth [3]. Studies have demonstrated that the source of CS, whether fungal or crustacean, can significantly influence physicochemical properties, such as M_w_, DDA, and mechanical strength [18]. Since the DDA and M_w_ of CS vary depending on the source, the material’s solubility, biocompatibility, and antimicrobial activity will also be affected.

Moreover, the biological properties of CS, such as biocompatibility, biodegradation, and antimicrobial activity, may also vary depending on its origin [9]. For instance, fungal-derived CS has been reported to exhibit lower immunogenicity compared to crustacean-derived CS, making it a more attractive option for specific tissue-engineering applications [9]. Additionally, the bioactivity of CS is critical in promoting osteogenic differentiation and mineralisation. CS’s bioactivity can also be influenced by M_w_, DDA, and origin (fungal- or crustacean-derived). The M_w_ of commercially available CS ranges from ~300 to 1000 kDa [19], with the DDA from 30% to 95% [20,21]. The DDA is the ratio between glucosamine and the sum of glucosamine and N-acetyl-glucosamine units [4]; thus, DDA corresponds to the free amino groups in the polysaccharide structure [22]. CS with a higher DDA value corresponds to a higher percentage of protonated primary amino groups, thus leading to an overall higher charge density [19]. For CT to be recognised as CS, the DDA required is >50% [21,23,24,25]. CS consists of β-(1→4) glycosidic linked D-glucosamine (deacetylated unit) and N-acetyl-D-glucosamine (acetylated unit) randomly distributed units [1,26,27]. CS scaffolds doped with other bioactive materials, such as hydroxyapatite, bioglass, and growth factors, enhance their osteogenic potential and further tailor their properties for BTE applications [3,28].

Understanding CS’s physicochemical and biological properties from different sources is crucial for developing effective BTE strategies. By characterising and optimising these properties, researchers can design CS-based scaffolds that promote bone regeneration and improve clinical outcomes for patients with bone defects. Here, we fabricated and characterised the physicochemical and biological properties of highly porous scaffolds synthesised from fungal- and crustacean-derived CS doped with different concentrations of tricalcium phosphate minerals (0, 10, 20 and 30(wt)%) to identify whether MDC is comparable to ADC as a more suitable variant for tissue-engineering applications. 

## 2. Materials and Methods

### 2.1. Fabrication and Sterilisation of Scaffolds

3(wt)% chitosan (Sigma-Aldrich, CAS: 9012-76-4, Taufkirchen, Germany, 3,100,000–3,750,000 Da, >75% deacetylated and, Chitolytic, C-M-98-501441) was dissolved in a 2(*v*/*v*)% acetic acid (Acros Organics, Geel, Belgium, MFCD00036152) solution and mixed via a magnetic stirrer for 24 h. The solution was then placed into an ultrasonic water bath for 30 min to allow for the removal of air bubbles [3]. After the stipulated time, different quantities of tricalcium phosphate mineral (TCP) (0, 10, 20 and 30(wt)%) were added to CS solutions and mixed for 1 h. The solutions were placed into 24-well plates and frozen at −80 °C for 24 h. Following freezing, the samples were placed into a freeze drier (VirTis 4 KB ZL Benchtop K, (SP Industries, Warminster, PA, USA) at −100 °C and 43 mTorr for 24 h. The lyophilised samples were subjected to a 15 min incubation with 1 M NaOH solution (Sigma-Aldrich, CAS: 1310-73-2, St. Louis, MO, USA) to attenuate the chitosan dissolution kinetics, then placed onto Whatman Grade 44 filtration paper (Merck, WHA1444110, Darmstadt, Germany) to eliminate superfluous NaOH residuals. The treated scaffolds were subjected to quintuple sequential washing using deionised water to ensure the complete removal of residual NaOH. A summary of the CS used is presented in Table 1, and details of the synthesised lyophilised scaffolds, including their respective code designations, are presented in Table 2.

### 2.2. Fourier-Transform Infrared Spectroscopy (FTIR)

The Vertex 70 FTIR spectrometer (Bruker, Billerica, MA, USA) was employed to perform molecular vibration spectroscopic analysis of the created scaffolds using the attenuated total reflection (ATR) mode. Scaffolds were scanned 200 times from 4000 cm^−1^ to 400 cm^−1^ at a spectral resolution of 4 cm^−1^. The light source and beam splitter were a MIR lamp and KBr, respectively.

### 2.3. Zeta Potential

The zeta potential of both unloaded and TCP mineral-loaded chitosan suspensions was determined by diluting the suspensions to concentrations of 2.9 g/dm^3^. Measurements were taken using Melvern Zetasizer equipment in cell DTS 1070 cuvettes. The refractive indices of CS and TCP minerals were 1.52 and 1.65, respectively.

### 2.4. Scanning Electron Microscopy (SEM)

The unloaded and TCP mineral-loaded scaffold morphology was studied using the Hitachi SU8230 1–30 kV (Hitachi High-Tech Corporation, Düsseldorf, Germany) cold field emission gun SEM. Before SEM, the samples were coated with 6 µm of gold to improve the electrical conductivity of the materials, enabling an improved signal-to-noise ratio. 

### 2.5. Scaffold Swelling and Degradation

All scaffolds were dried in a furnace at 50 °C for 5 h and then weighed (*W_d_*) before the swelling experiment. The scaffolds (*n* = 3 of each type) were submerged into phosphate-buffered saline (PBS) (Lonza, catalogue: BE17-517Q, Basel, Switzerland) for 6 h. The scaffolds were removed from PBS and re-weighed using an electronic balance every hour. The swelling % of both groups of scaffolds was calculated using the following Equation:*Swelling % = (**W**_w_ − **W**_d_)/**W**_d_* × 100(1)

***W_w_*** and ***W****_d_* are the wet and dry weights of the samples, respectively. The scaffold degradation (*n* = 3) capabilities were assessed weekly for 8 weeks in PBS. At each time point, the scaffolds were removed from PBS solutions, dried at 50 °C for 24 h, and re-weighed at each time point. The scaffold’s weight losses were calculated using Equation (2):Δ***W****_d_ (%) = (**W***_0_
*− **W**_d_*_1_*)/**W***_0_ × 100(2)

***W***_0_ and ***W****_d_*_1_ refer to the initial scaffold weights and the scaffold weights at time (*t*), respectively. 

### 2.6. Ethics, Sample Processing and Cell Culture

Ethical approval was obtained from the Yorkshire and Humberside National Ethics Research Committee (ethics number 06/Q1206/127) to collect human bone marrow aspirate (BMA) samples to generate mesenchymal stromal cells (BMMSCs). As previously described, the BMA was collected in ethylenediaminetetraacetic acid (EDTA) tubes [29]. The BMA samples were first passed through sterile 70 μm cell strainers (Falcon, Fisher Scientific, Loughborough, UK) to exclude fat or bone debris. Then, it was treated with ammonium chloride to lyse erythroid lineage cells, as described previously [30] and cultured in StemMacs (SM) MSC expansion media (Miltenyi Biotec, Bisley, UK) containing 1% (*v*/*v*) penicillin-streptomycin (PS) for three passages. The cells were then frozen at −80 °C for further use. BMMSCs from three donors (*n* = 3) aged 30 to 50 with high cellularity were defrosted and pooled to perform several cell culture experiments.

### 2.7. Sterility Testing 

The scaffolds were sterilised by immersion in 70% ethanol for 2 min, followed by three PBS washes to ensure the removal of ethanol traces. The scaffolds were primed for experiments by placing into 2 mL of SM containing 1% PS within 24-well plates, then incubated at 5% CO_2_ and 37 °C for 1 week. After 7 days, the scaffolds were placed into 6-well plates containing 50% confluent monolayers of pooled BMMSCs and imaged up to day 7 to examine the direct or contact cytotoxicity.

### 2.8. Contact Cytotoxicity Assay by Giemsa Staining

Contact testing was conducted per ISO10993-5:2009 Part 5: Tests for in vitro cytotoxicity [31]. The scaffolds (*n* = 3) were placed in 6-well plates. The positive control consisted of cells without scaffolds, while the negative control used 40% dimethyl sulfoxide (DMSO). The wells were washed twice with Dulbecco’s phosphate-buffered saline (DPBS) and then aspirated. Next, 2 mL of cell suspension containing 1 × 10^4^ cells was added to each well. The plates were incubated at 37 °C with 5% CO_2_ for 7 days. After 7 days, the media were aspirated, and the wells were washed twice with DPBS. Each well received 1 mL of 4% neutral-buffered formalin (NBF) and was incubated for 15 min. The formalin was aspirated, and the wells were stained with Giemsa solution for 5 min. The wells were then washed with distilled water and air-dried for 24 h. The samples were examined microscopically using a Leica DM16000 B inverted microscope to record any changes in cell morphology, confluency, attachment, and detachment. All images were collected digitally. The wells were graded as per Table 3.

### 2.9. Indirect Toxicity—Cytotoxicity and Proliferation via XTT Assay

The XTT assay kit (Roche, Mannheim, Germany) measures the potential cytotoxicity of scaffolds by assessing the decrease in living cell numbers corresponding to a reduction in the overall activity of mitochondrial dehydrogenases. Thus, the reduction is directly linked to the formation of orange formazan, as observed through optical density measurements at 450 nm. Sterilised scaffolds were placed in duplicates in 6-well plates containing 5 mL of SM media and incubated at 37 °C for a period spanning from 72 h to up to 2 months (ISO 10993:2021 part 12) [32]. The elutes were collected on the harvesting time points, i.e., 3 days, 1, 2, 3, 4 and 8 weeks, labelled, and stored in 620 µL portions at −80 °C until the cytotoxicity evaluation was performed in accordance with ISO 10993-5:2009(E) part 5 [31].

For indirect cytotoxicity evaluation, BMMSCs from three different cultures (*n* = 3) were pooled for feasibility and timely completion of the investigation. The pooled culture was placed in triplicate in 96-well plates containing 200 µL of SM media with a cell density of 10,000 cells/well and incubated for 24 h. The basal media was then replaced with 200 µL of defrosted treatment media containing scaffold eluate, negative control (SM with 10% DMSO), or positive control (SM media). The cells were then incubated in the treatment media for 24 h before the addition of the XTT reagents as per the manufacturer’s instructions. After treatment, the wells were aspirated and treated with 100 μL of DMEM containing 10 (*v*/*v*)% FCS (both from Thermofisher Scientific, Loughborough, UK) and 50 µL of the XTT solution, followed by a 4 h incubation at 37 °C. The plates were read at the corresponding reference wavelengths of 450 nm and 630 nm. The 630 nm values were subtracted from the 450 nm values to obtain the final optical densities (OD). Test well ODs were normalised to positive controls to evaluate cell viability or inhibition of proliferation.

The pooled BMMSCs (*n* = 3) for proliferation analysis were seeded at 500 cells/well density in a 96-well plate for 24 h in SM media. After 24 h, the basal media were replaced with 200 µL of treatment media containing either scaffold eluate, negative control (SM media with 10 (*v*/*v*)% DMSO), or positive control (SM media). Cells were then cultured in the treatment media for 4 days. Following this period, the XTT assay was performed as previously described, and the results were analysed to determine cell proliferation relative to the positive control.

### 2.10. Fluorescence Actin and Nuclei Staining

Cell adhesion was observed by placing 4 × 10^4^ BMMSC cells on freeze-dried, unloaded, and TCP mineral-loaded scaffolds for 48 h. After the designated time, the scaffolds were rinsed twice with PBS, fixed using 10% neutral-buffered formalin (NBF), permeabilised with 1 (*v*/*v*)% Triton x-100 (Sigma-Aldrich) for 5 min, and washed twice with PBS. The actin filaments of BMMSC cells were stained with Alexa Fluor-488 phalloidin (Invitrogen, Waltham, MA, USA) for 2 h, and the cell nuclei were stained using 4′,6-diamidino-2-phenylindole (DAPI) dye (Sigma-Aldrich) for 15 min. The scaffolds were washed twice with PBS and visualised using a Leica TCS SP8 confocal microscope.

### 2.11. Statistics

Data are shown as mean ± standard deviation. Differences between groups were analysed using two-way ANOVA. Statistical evaluation and graphic illustrations of the data were performed using GraphPad Prism (version 9.2.0). A *p*-value of <0.05 was deemed statistically significant.

## 3. Results

### 3.1. Fourier-Transform Infrared Spectroscopy (FTIR)

FTIR analysis (Figure 1) identified differences in the molecular structure and interactions between the ADC and MDC scaffolds embedded with TCP minerals of varying concentrations (0, 10, 20 and 30(wt)%). Although both types of CS scaffolds contain the same molecular bonds, the interaction between CS and TCP in the samples causes shifts in peak positions, broadening/sharpening peaks, and changes in peak intensities. Other researchers observed similar results [3,33,34,35]. The specific FTIR peaks observed in the CS scaffolds vary depending on the preparation method, mineral particle size, impurities and the concentration of the TCP mineral in the samples. The presence of both CS and TCP leads to overlapping peaks; CS typically shows characteristic bands related to O-H and N-H stretching (3200–3500 cm^−1^), amide I (1620–1650 cm^−1^), amide II (1550–1590 cm^−1^), and C-O stretching corresponding to peaks in the 1000–1150 cm^−1^ region. TCP minerals exhibit bands in 1000–1100 cm^−1^ (υ_3_), 950–970 cm^−1^ (υ_1_), 400–450 cm^−1^ (υ_2_), and 550–600 cm^−1^ associated with (υ_1_) bending mode. Other spectral features often overshadow the υ_1_ as this band is less intense. The position and intensity of the bands are known to vary depending on the minerals’ crystallinity and the interactions between the TCP and CS polysaccharide matrix. For MDC and ADC scaffolds, the results indicate that the addition of TCP mineral content from 0 to 30(wt)% caused a corresponding increase in intensity in the TCP spectral bands, which is to be expected. Table 4 summarises the specific molecular bonds related to CS and TCP minerals.

### 3.2. Zeta Potential

The zeta potential trend presented in Figure 2 for freeze-dried CS scaffolds, synthesised from fungal and crustacean sources, reveals a progressive reduction in positive charge as the TCP mineral content increases. CS is a cationic polymer; therefore, its charge density is influenced by the number of protonated amino groups (NH_3_^+^). The decline in zeta potential can be ascribed to the electrostatic interactions between the positively charged NH_3_^+^ groups in CS and the negatively charged PO_4_^3−^ groups inherent to TCP minerals. The findings confirm that scaffolds devoid of TCP minerals exhibited a more pronounced positive charge compared to those containing minerals, whereby MDC-1 and ADC-1 scaffolds lacking TCP minerals presented the highest zeta potentials of 47.3 ± 1.2 mV and 55.1 ± 1.6 mV, respectively. A more significant positive charge influences scaffold properties, such as stability, dispersibility, and interaction with surrounding biological components, particularly the scaffold’s performance in biocompatibility, bioactivity, and cellular interactions. 

### 3.3. Scanning Electron Microscopy

Figure 3 presents scanning electron microscopy (SEM) micrographs of fungal- and crustacean-derived CS scaffolds incorporating various concentrations (0, 10, 20, and 30(wt)%) of TCP minerals. A prominent distinction between the two CS-derived scaffolds is the striation-like appearance of MDC structures, in contrast to the more groove-like porous morphology exhibited by ADC scaffolds. The MDC scaffold structure became tighter and more defined as TCP content increased. Specifically, MDC-1 (0% TCP) presented wide striations averaging 10–15 µm in width, while MDC-2 (10% TCP) showed slightly tighter striations averaging 8–12 µm. MDC-3 (20% TCP) depicts tighter striations, averaging 5–10 µm, and MDC-4 (30% TCP) displayed the tightest striations, averaging 3–8 µm, reflecting a significant interaction between chitosan and TCP particles. In contrast, ADC scaffolds exhibited groove-like porous structures with pore size and configuration influenced by TCP content. ADC-1 (0% TCP) had a baseline porous structure with pore sizes ranging from 20 to 30 µm, ADC-2 (10% TCP) featured more and larger pores, ranging from 25 to 35 µm, ADC-3 (20% TCP) had interconnected pores forming a network-like structure with pores averaging 30–40 µm, and ADC-4 (30% TCP) shows pore coalescence with sizes between 35 and 45 µm. The differences in the surface structure of MDC and ADC influence cell adhesion and proliferation.

### 3.4. Scaffold Swelling and Degradation

The hydrodynamic characteristics of lyophilised CS scaffolds, incorporating varying concentrations of TCP minerals, are illustrated in Figure 4a,b, delineating their temporal evolution. A noticeable initial escalation in swelling ratios is apparent for MDC and ADC scaffold variants within the 0–15 min interval. Subsequently, a moderate yet sustained increment occurs between the 15 and 180 min timeframes, concluding in mass equilibration in the 180 to 360 min window. Upon completion of the 6 h period, the swelling percentages of the scaffolds were ascertained as follows: for MDC-1, MDC-2, MDC-3, and MDC-4, the values were 153.63 ± 7.6%, 133.87 ± 5.6%, 127.46 ± 4.5%, and 113.60 ± 6.4%, respectively. In terms of, ADC-1, ADC-2, ADC-3, and ADC-4 demonstrated swelling percentages of 111.5 ± 6.8%, 104.8 ± 3.5%, 100.1 ± 4.3%, and 88.03 ± 5.7%, respectively.

Elevating the TCP mineral content decreased mass degradation for the scaffolds, with MDC-4 and ADC-4 scaffolds demonstrating the most minimal mass reduction compared to the lyophilised scaffolds devoid of minerals. The data presented in Figure 4c reveals that the MDC scaffolds experienced a more substantial mass degradation overall among all scaffold categories relative to the ADC scaffolds (Figure 4d). At the conclusion of the eighth week, mass reductions of 55.98 ± 4.2%, 40.16 ± 3.6%, 27.05 ± 4.7%, and 19.16 ± 5.3% were observed for MDC-1, MDC-2, MDC-3, and MDC-4, respectively, while ADC-1, ADC-2, ADC-3, and ADC-4 exhibited mass reductions of 35.78 ± 5.1%, 25.19 ± 4.2%, 20.23 ± 6.3%, and 13.68 ± 5.4%, respectively. The swelling % and the mass degradation are overall higher for MDC scaffolds than ADC, with an increasing degradation rate over time, particularly for MDC-1 (see Appendix A). The lowest degradation rate was observed for ADC-4 scaffolds containing 30(wt)% TCP minerals. The addition of TCP minerals influenced the microstructure, as evidenced by SEM characterisations, and impacted the overall charge profile, swelling, and degradation properties. Subsequently, we explored how these variations influenced the scaffolds’ cytocompatibility and cell attachment properties, conducting in vitro experiments using BMMSCs.

### 3.5. Sterility Testing and Direct Toxicity 

A sterility assessment was performed by immersing the scaffolds in SM for one week. Systematic observations and high-resolution imaging of the media and scaffolds were executed on days 1, 3, and 7. Additionally, scaffold imaging was conducted, and representative images are provided in Appendix A. No discernible alterations in media chromaticity were observed and the absence of turbidity or microbial contamination in the wells was confirmed throughout the entire week, confirming that the preconditioned scaffolds effectively retained sterility in the culture medium until day 7. Once sterility was confirmed, the scaffolds’ cytotoxicity was examined. BMMSCs were seeded on 6-well plates in SM media to evaluate direct cytotoxicity. Preconditioned scaffolds were incorporated into the plates upon reaching 50% cellular confluency. High-resolution imaging of the cell-scaffold interface was executed to investigate the immediate impact of cell-scaffold interactions, a process referred to as direct cytotoxicity assessment. High cell viability was observed in MDC and ADC relative to the control group. BMMSCs within MDC-1 and MDC-2 had higher cell viability on days 3 and 7, whereas MDC-3 and MDC-4 exhibited diminished survival rates, particularly on day 7 (see Appendix A). Analogous findings were observed in the crustacean-derived chitosan (ADC) scaffolds, suggesting that increasing TCP mineral concentrations for both CS types reduced cell proliferation.

### 3.6. Contact Cytotoxicity Assay by Giemsa Staining

Contact cytotoxicity testing is a qualitative assessment of cytotoxicity via microscopic observations to determine any changes to the cells’ morphology and reactivity zones undertaken per ISO10993-5:2009 [31]. Based on the Giemsa-stained cells shown in Figure 5, it is evident that the scaffolds, whether ADC or MDC, exhibited cytotoxic effects on BMMSC cells after a 7-day growth period. The images display healthy, intact cells with typical morphology, confluency, and attachment across all scaffold types and TCP concentrations compared to the control images. The absence of cell death or detachment highlights the non-toxic nature of all scaffold materials; the results are an indication that both ADC and MDC scaffolds support cell viability and do not induce any toxic responses in BMMSC cultures. Therefore, the CS’s (MDC and ADC) were graded as 0 concerning ISO10993-5:2009 [31], whereby the scaffolds displayed “no detectable zone around or under specimen”.

### 3.7. Indirect Toxicity—Cytotoxicity and Proliferation by XTT

Indirect cytotoxicity was assessed by applying the XTT assay in a 96-well plate configuration, subsequent to 24 h cell plating and exposure to culture media containing scaffold eluates for up to two months. As delineated in Figure 6, cellular viability in response to media eluates across all time points exhibited >80% survival compared to the negative control, while, in some cases, it was greater than the positive control. Comparing ADC and MDC scaffolds, the % cell viability across all time points was >80%. The overall trend depicts a slight decline in cell viability as time progressed, whereby compared to ADC scaffolds, the MDC scaffolds presented reduced cell viability from week 2 onwards, with significant differences observed between weeks 1 and 4. No significant differences were observed for scaffolds containing TCP minerals. Based on the 24 h cytotoxicity data, ADC/ADC-1 scaffolds seemed to present elevated cell survival compared to the MDC/MDC-1 scaffolds. Increasing TCP mineral content, particularly for MDC-3/MDC-4 scaffolds, increased cell survival in comparison to ADC-3/ADC-4 scaffolds.

An analogous methodology was used for assessing cell proliferation, where the cells were exposed to media eluates for 96 h to evaluate the proliferation rate of BMMSCs under the influence of the media eluates. Figure 7 demonstrates that the % cellular proliferation in the presence of media eluates from all time points surpassed the negative controls. Interestingly, MDC-1 indicated a trend for being more robust for BMMSC with the highest % cell survival for MDC-2, MDC-3 and MDC-4 compared to their ADC counterparts (Figure 7). In the context of MDC scaffolds, MDC-1, MDC-3, and MDC-4 scaffolds indicate higher cell viability in comparison to ADC scaffolds, albeit non-significant. Interestingly, MDC-2 demonstrated lower % cell viability via proliferation compared to ADC-2, even though it was non-significant. Although the data were not statistically significant, this further implies that MDC and ADC scaffolds effectively promoted cell survival and proliferation, surpassing the performance of the negative control closely approximating the positive controls.

### 3.8. Fluorescence Actin and Nuclei Staining

Fluorescence staining with Alexa Fluor-488 and DAPI was employed to visualise and analyse cellular interactions within the freeze-dried CS scaffolds embedded with TCP minerals (Figure 8). Alexa Fluor-488, a green fluorophore, was utilised to stain actin filaments and other cellular components, while DAPI, a blue fluorophore, was employed to label cell nuclei. The dual-staining approach facilitated the evaluation of cellular adhesion, proliferation, and cell distribution within the TCP-mineral-embedded CS scaffolds. The fluorescence staining experiments revealed suboptimal cell attachment to the scaffold surfaces without initial SM scaffold priming with increasing TCP mineral content, particularly in ADC scaffolds. Therefore, scaffold priming was required to increase the probability of BMMSCs attaching to the freeze-dried scaffolds. Upon comparing the morphological characteristics of BMMSCs on MDC and ADC scaffolds, we observed improved attachment on MDC scaffolds, particularly for the MDC-2 scaffold variant. Here, the cells demonstrated favourable morphology, with well-spread cytoskeletal structures extending across the scaffold surface, indicative of a healthy and well-adapted cellular state, as depicted in Figure 8b. 

## 4. Discussion

We chose high M_W_ CS from animal and fungal sources for our comparison for several key reasons. Specifically, we investigated commercially available crustacean-derived CS (ADC) with an M_W_ of 330–375 kDa and fungal-derived chitosan (MDC) with an M_W_ of 200–300 kDa [40]. The M_W_ significantly influences CS behaviour, including swelling, zeta potential, and cell interactions. High-molecular-weight CSs are known for their superior mechanical strength, film-forming ability, and biocompatibility, making them ideal for various industrial and research purposes [41]. Our selection reflects real-world applications and the availability of CS on the market, ensuring the relevance and applicability of our findings. Researching commercially available forms of CS allows for easier replication and application of our results by other investigators. By comparing high-molecular-weight CS from different sources, we aimed to provide insights into how the origin affects the physicochemical and biological properties, ensuring our study is scientifically and practically relevant.

Comparative FTIR studies in our investigation revealed subtle differences in the spectra of ADC and MDC freeze-dried scaffolds, particularly when doped with varying TCP mineral concentrations. Notably, the MDC scaffold samples exhibited slightly broader and more pronounced peaks, likely attributed to increased DDA and decreased extraction and processing impurities in contrast to ADC-derived scaffolds [42]. CS obtained from crustacean sources often contains additional mineral impurities due to the extraction and harsh processing conditions. These impurities can cause minor shifts in peak positions and variations in peak intensities within the FTIR spectrum. In contrast, fungal-derived CS is typically free from impurities and proteins, such as tropomyosin, arginine kinase and myosin light chain present in CS derived from crustacean sources and are known allergens [9]. The absence of these harmful trace contaminants, coupled with fewer processing steps, enables the production of high-purity CS [43]. Fungal CS has been reported to not readily cause allergic reactions, toxicity, or inflammation once implanted into the body [7,22], thus enhancing it as a material for tissue engineering. Furthermore, CS derived from fungal sources exhibits distinct characteristics, including but not limited to different M_w_s, DDAs, and variations in the distribution of charged groups [44]. Although fungal-derived CS has ecological and non-allergic advantages, the limiting factor seems related to extraction and processing techniques, resulting in lower yields than those of crustacean-derived CS [16].

Tricalcium phosphate (TCP) minerals were chosen over other elements, such as hydroxyapatite (HAP) or growth factors, due to TCP’s ability to degrade at a controlled rate in the body, releasing calcium and phosphate ions essential for bone metabolism and remodelling. The gradual resorption aligns well with the natural bone-healing process, whereas HAP is less resorbable and may persist longer in the body, potentially leading to complications [45]. TCP is relatively cost-effective and widely available compared to some growth factors, which can be expensive and difficult to produce in large quantities. While growth factors such as BMPs (Bone Morphogenetic Proteins) are highly effective in promoting bone growth, their use can be associated with risks such as ectopic bone formation or inflammation [46,47,48]. To minimise these risks while promoting effective bone regeneration, we found TCP to be a more practical choice for scalable and cost-effective BTE applications; this selection ensured a balance between efficacy and safety, making TCP an ideal candidate for our scaffold designs.

The presence of TCP in both MDC and ADC scaffolds led to the formation of hydrogen bonds between the protonated amino groups of CS and the phosphate groups of TCP, contributing to a stable composite structure [7,49]. The polycationic nature of CS plays a vital role in BTE applications, as the formation of polyelectrolyte complexes can be produced with the anionic biological macromolecules [50,51], including but not limited to lipids, minerals, proteins, DNA and polymers, i.e., poly(acrylic acid) [4,52,53]. Crosslinking CS scaffolds with materials containing at least two reactive functional groups, i.e., calcium phosphates, composites (nano-zirconia and nano-calcium zirconate) and bioglass, increased the overall scaffolds’ mechanical properties when compared to complexes solely containing CS [54]. Crosslinking reduces the protonated amino groups, forming bridges between the CS polymeric chains, thus leading to structural stabilisation [3,7].

The findings, particularly regarding the zeta potential between MDC and ADC CS scaffolds, offer new insights into scaffold design for bone regeneration. The higher zeta potential observed in MDC scaffolds suggests a more reactive surface, which could enhance osteoconductive properties, consistent with the notion that surface charge significantly influences protein adsorption and cell behaviour [43]. The zeta potential of CS is contingent upon the M_w_. High-M_w_ CS possesses a more extensive polymeric chain structure, indicating an increased presence of functional groups than chitosan exhibiting lower M_w_. Consequently, the relative positive charge (+ve) is directly proportional to the abundance of protonated amino groups within the structure. As anticipated, the MDC-1 and ADC-1 mineral-free scaffolds exhibited the highest zeta potential values of 47.3 ± 1.2 mV and 55.1 ± 1.6 Mv. The zeta potentials of the synthesised freeze-dried scaffolds displayed a decreasing trend, where an escalation in TCP concentration resulted in a decline in the positive zeta potential values. The reduction in zeta potential is attributable to the increase in TCP phosphate ions, which generate ionic bonds or electrostatic interactions with the protonated amino groups in CS [3]. These observations are in accordance with findings documented by other researchers in the field [55]. 

Microstructural differences were observed between the two types of CS scaffolds via SEM (Figure 3), particularly for increasing TCP mineral content. The impact of TCP mineral content on MDC and ADC scaffolds with escalating TCP mineral concentrations modulated the morphological properties of both scaffold types. The MDC scaffolds manifested a striation-like appearance, whereas ADC scaffolds demonstrated a more porous morphology. The variation in microstructure is pivotal in determining the suitability of these scaffolds for specific BTE applications, as scaffold architecture plays a crucial role in cell migration, cell adhesion, proliferation, nutrient diffusion, and overall tissue integration [2]. It has been reported that CS scaffolds prepared from fungal sources presented increased porosity and demonstrated higher thickness, opacity, liquid uptake and water permeability alongside being non-toxic to fibroblast cells [43]. The striation-like structure of MDC scaffolds may facilitate increased cell alignment and tissue organisation. In contrast, the porous nature of ADC scaffolds could be more beneficial for vascularisation and nutrient transport, aligning with findings from [56].

CS is a hydrophilic biopolymer [57] that facilitates the diffusion of water molecules due to the structural free volume and the ease of polymer chain mobility [58,59]. Our findings demonstrate a distinct pattern in the swelling behaviour of MDC and ADC scaffold variants, with an initial rapid increase in the swelling ratios within the first 15 min, followed by a gradual increase up to 360 min. Notably, scaffolds with higher TCP content exhibited lower swelling and degradation percentages, attributed to the stabilisation of CS biopolymer chains due to the interaction of the CS protonated amino and the TCP phosphate groups. DDA and M_w_ are inversely proportional to CS’s swelling capacity and degradation [1,60,61]. Increased DDA correlates to increased crystallinity, reducing the swelling index [60,61] and degradation rates [19,62]. The DDA plays a vital role in biological in vitro and in vivo degradation, healing capacity, osteogenesis and lysozyme degradation within biological systems [63,64]. CS with a high DDA, i.e., 84% to 90%, exhibits delayed degradation and presents a lower degradation index than CS DDA between 65% and 82% [7]. CS containing high levels of DDA have been shown to degrade slowly over several months [63,64], which is beneficial for bone regeneration as the degradation rate can be tuned to match bone regrowth [19]. Low M_w_ CS contains smaller polysaccharide chains, reducing chain entanglements [65]. Therefore, the smaller CS polysaccharide chains degrade more rapidly into variable-length oligosaccharides than CS with higher M_w_ [61,66]. The degradation difference between low- and high-CS DDA is related to increased crystallinity hydrogen bonding [7,63,64]. Under physiological conditions, CS provides a controlled chemical breakdown leading to inert degradation products, including non-toxic oligosaccharides [67], N-acetyl-d-glucosamine residue and water, which can be utilised in metabolic pathways or excreted [6]. 

CS expresses a range of favourable properties, such as biodegradability [3,68], antibacterial, antifungal [69], minimal toxicity, biocompatibility [70], and wound-healing capability [71]. In contrast to ADC, MDC is not subject to seasonal or geographical limitations and does not require harsh acid treatments for purification and demineralisation to remove calcium carbonate and other minerals. Fungal-derived CS integrates a more malleable branched β-glucan, resulting in an inherent nanocomposite architecture that facilitates the formation of robust and tenacious fibre networks upon processing [72]. We observed significant differences between MDC-1 and ADC-1 scaffolds concerning cell viability for cytotoxicity; however, this was observed only in the scaffolds containing no TCP mineral content. Our data indicates MDC scaffolds can be an alternative to traditional crustacean-derived CS-based biomaterials. Bone regeneration [7,68] has been observed for CS scaffolds combined with osteogenic minerals such as hydroxyapatite (HAP) [73,74]. Osteogenesis is promoted for CS scaffolds containing immobilised adhesive peptides, e.g., tri-amino acid sequence arginine–glycine–aspartate [75,76]. CS supports cell attachments and the proliferation of osteoblast cells, leading to the formation of in vitro mineralised bone matrix [19,77]. Furthermore, as confirmed via micro-computed tomography, CS scaffold composites containing nano-HAP observed an increase in osteoblast cell proliferation after 8 weeks [78]. 

## 5. Conclusions

Our findings verify previous research highlighting the significant influence of scaffold composition and surface characteristics on cellular behaviour. The incorporation of TCP minerals in ADC and MDC scaffolds led to observable structural differences, as revealed through SEM analysis. Notably, the striated structure of MDC scaffolds seemed to enhance cell attachment compared to the more conventional pore structure of ADC scaffolds, particularly for MDC scaffolds containing 10(wt)% TCP (i.e., MDC-2). CS scaffolds from crustacean and fungal sources exhibited similar cellular toxicity profiles. However, we acknowledge certain limitations in our study; for example, the experiments were conducted in vitro using pooled BMMSCs from three donors to ensure the feasibility and timely completion of the project. Pooling cells minimises the variability often seen in primary cells from different donors. It is important to note that our study focused exclusively on BMMSCs, the progenitors of bone cells. Therefore, further research is needed to investigate the effects on early-stage osteoblasts and mature osteocytes. Understanding how different cell types interact with CS scaffolds is crucial for expanding their applications in BTE. Future research should (i) compare the in vitro and in vivo applications of MDC and ADC scaffolds to enhance our understanding of CS from various sources, aiming to reduce immunogenic reactions and develop more patient-friendly scaffolds for bone regeneration; (ii) focus on isolating and comparing chitosan fragments with similar molecular weights, conducting in vivo studies, and investigating interactions with different cell types to broaden the applicability and understanding of our scaffolds; and (iii) explore the mechanistic pathways through which molecular weight influences scaffold properties and cellular interactions.

## Figures and Tables

**Figure 1 bioengineering-11-00720-f001:**
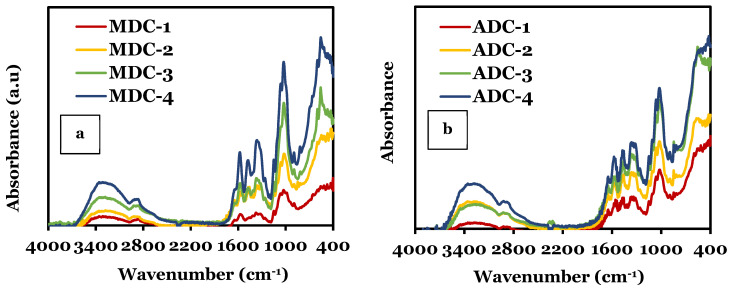
Comparison of Fourier-transform infrared spectroscopy spectra of synthesised freeze-dried chitosan (CS) scaffolds containing varying concentrations of tricalcium phosphate (TCP) minerals (0, 10, 20 and 30(wt)%) data obtained in the 4000 cm^−1^ to 400 cm^−1^ regions at a resolution of 4 cm^−1^ and using the Vertex 70 FTIR spectrometer, USA, in attenuated total reflection mode: (**a**) fungal-derived CS scaffolds (MDC), (**b**) crustacean-derived CS scaffolds (ADC).

**Figure 2 bioengineering-11-00720-f002:**
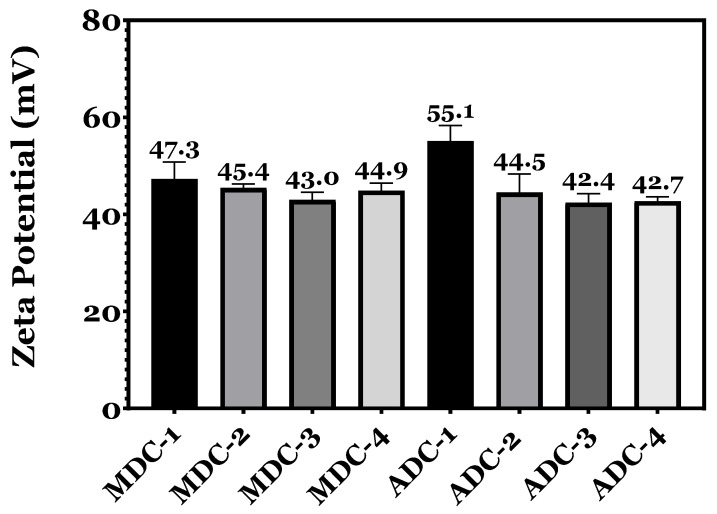
Zeta potential measurements of unloaded and tricalcium phosphate (TCP) mineral-loaded freeze-dried chitosan scaffold suspensions derived from fungal (MDC) and crustacean (ADC) sources were conducted using a Malvern Zetasizer. The concentrations of TCP incorporated into the scaffolds were 0(wt)% (MDC-1 and ADC-1), 10(wt)% (MDC-2 and ADC-2), 20(wt)% (MDC-3 and ADC-3), and 30(wt)% (MDC-4 and ADC-4).

**Figure 3 bioengineering-11-00720-f003:**
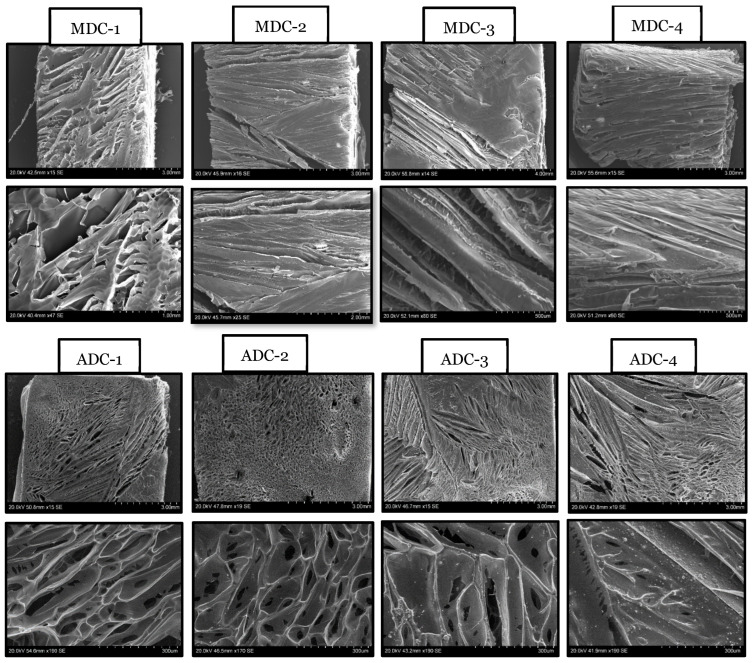
Hitachi S-3400N scanning electron microscopy micrographs of the freeze-dried fungal- (MDC, **top panel**) and crustacean (ADC, **bottom panel**)-derived chitosan scaffolds embedded with varying concentrations of TCP incorporated into the scaffolds were 0(wt)% (MDC-1 and ADC-1), 10(wt)% (MDC-2 and ADC-2), 20(wt)% (MDC-3 and ADC-3), and 30(wt)% (MDC-4 and ADC-4).

**Figure 4 bioengineering-11-00720-f004:**
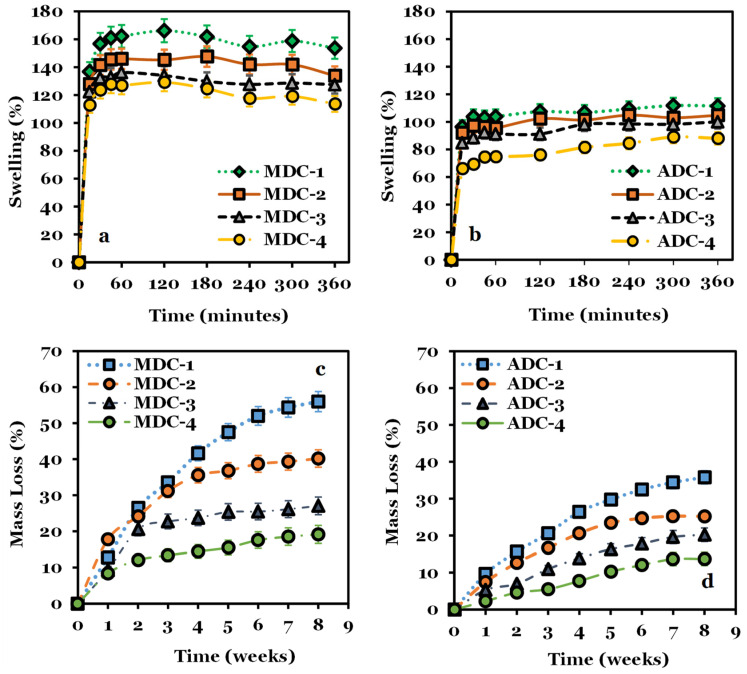
(**a**,**b**) Display the swelling kinetics and (**c**,**d**) degradation profiles of lyophilised chitosan scaffolds derived from both fungal (MDC) and crustacean (ADC) sources, incorporating various concentrations of tricalcium phosphate (TCP) minerals. The scaffolds were immersed in phosphate-buffered saline (pH 7.4) at a physiological temperature of 37 °C. MDC and ADC scaffolds were fabricated with 0(wt)% (MDC-1 and ADC-1), 10(wt)% (MDC-2 and ADC-2), 20(wt)% (MDC-3 and ADC-3), and 30(wt)% (MDC-4 and ADC-4) TCP concentrations.

**Figure 5 bioengineering-11-00720-f005:**
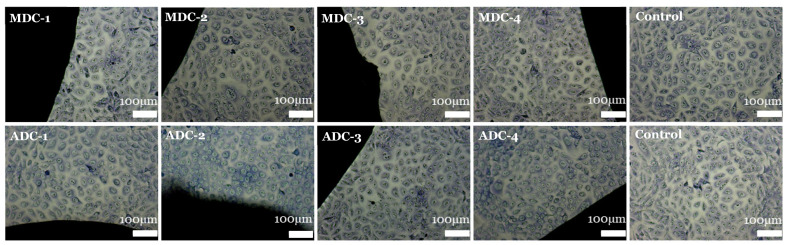
Contact toxicity via Giemsa assay for fungal- (MDC) and crustacean (ADC)-derived chitosan scaffolds doped with different concentrations of tricalcium phosphates minerals (0(wt)% (MDC-1), 10(wt)% (MDC-2), 20(wt)% (MDC-3) and 30(wt)% (MDC-4)). Compared to the control group (absence of scaffold), the cellular morphology near the scaffolds is visualised using the Leica DM16000 B inverted microscope.

**Figure 6 bioengineering-11-00720-f006:**
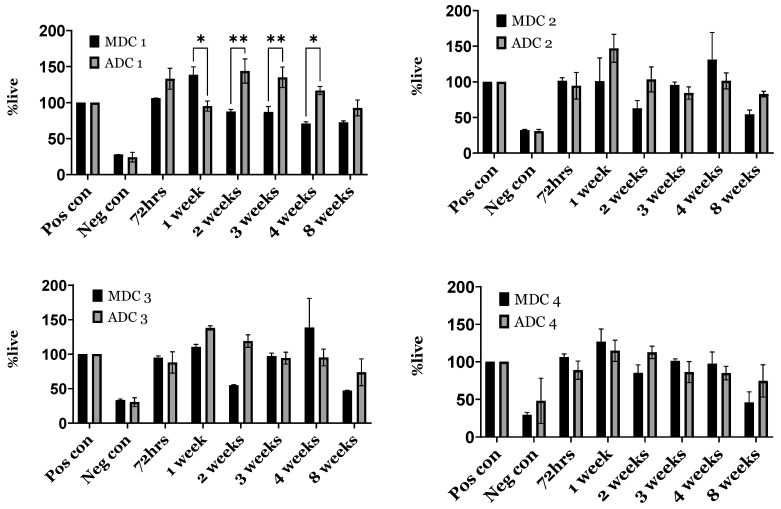
Indirect cytotoxicity evaluation (XTT assay) of bone-marrow-derived mesenchymal stromal cells (BMMSCs) cultured with fungal-derived chitosan (MDC) and crustacean-derived chitosan (ADC) scaffolds with varying tricalcium phosphate (TCP) content. MDC and ADC scaffolds were prepared with 0(wt)% (MDC-1 and ADC-1), 10(wt)% (MDC-2 and ADC-2), 20(wt)% (MDC-3 and ADC-3), and 30(wt)% (MDC-4 and ADC-4) TCP. The cell cytotoxicity across all ADC and MDC scaffold formulations was normalised to the positive control. * *p* < 0.05, ** *p* < 0.01 denote levels of statistical significance.

**Figure 7 bioengineering-11-00720-f007:**
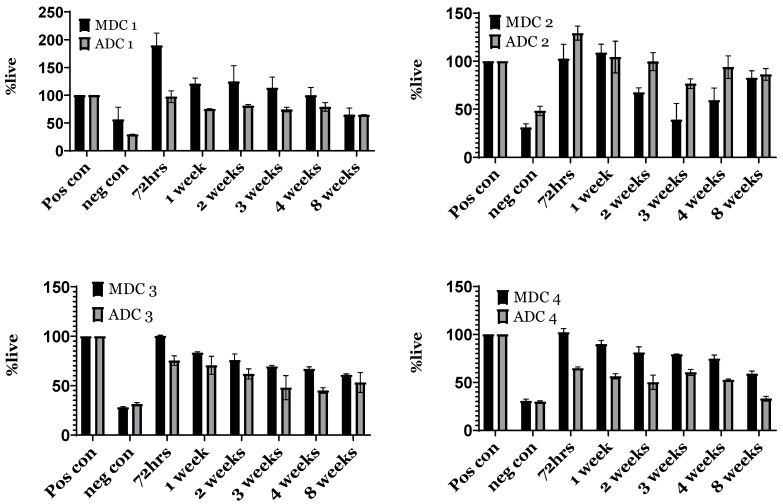
Evaluation of proliferation using the XTT assay for bone-marrow-derived mesenchymal stromal cells (BMMSCs) cultured with fungal-derived chitosan (MDC) and crustacean-derived chitosan (ADC) scaffolds with varying tricalcium phosphate (TCP) content. MDC and ADC scaffolds were prepared with 0(wt)% (MDC-1 and ADC-1), 10(wt)% (MDC-2 and ADC-2), 20(wt)% (MDC-3 and ADC-3), and 30(wt)% (MDC-4 and ADC-4) TCP, The cell proliferation across all ADC and MDC scaffold formulations was normalised to the positive control.

**Figure 8 bioengineering-11-00720-f008:**
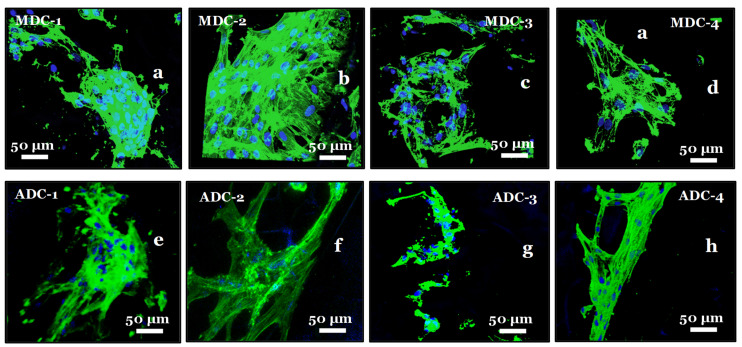
Fluorescence microscopy images of BMMSCs cultured on fungal-derived chitosan (MDC) and crustacean-derived chitosan (ADC) scaffolds with varying calcium phosphate (TCP) content. MDC and ADC scaffolds were prepared with 0(wt)% ((**a**) MDC-1 and (**e**) ADC-1), 10(wt)% ((**b**) MDC-2 and (**f**) ADC-2), 20(wt)% ((**c**) MDC-3 and (**g**) ADC-3), and 30(wt)% ((**d**) MDC-4 and (**h**) ADC-4) TCP. Cells were stained with Alexa Fluor-488 (green) to visualise actin filaments and DAPI (blue) to label cell nuclei.

**Table 1 bioengineering-11-00720-t001:** Structural differences between crustacean- and fungal-derived chitosan.

Chitosan	Code	Molecular Weight (kDa)	Viscosity (mPa.s/cps)	Degree of Deacetylation (%)
Fungal	MDC *	200–300	600	98.1
Crustacean	ADC **	330–375	2000	≥75

*** MDC** refers to fungal/mushroom-derived chitosan (high-molecular-weight chitosan, Chitolytic, C-M-98-501441) **** ADC** refers to crustacean-derived chitosan (high-molecular-weight chitosan, Sigma Aldrich, CAS: 9012-76-4).

**Table 2 bioengineering-11-00720-t002:** Fabricated scaffolds with corresponding code names.

Sample Code	Description	TCP (wt)%
TCP	Tricalcium Phosphate Mineral	-
MDC-1/ADC-1	Freeze-dried Chitosan Scaffold	0
MDC-2/ADC-2	10(wt)% TCP mineral loaded chitosan scaffold	10
MDC-3/ADC-3	20(wt)% TCP mineral loaded chitosan scaffold	20
MDC-4/ADC-4	30(wt)% TCP mineral loaded chitosan scaffold	30

**Table 3 bioengineering-11-00720-t003:** Reactivity grades for contact cytotoxicity testing.

Grade Reactivity	Description
0	None: No detectable zone around or under specimen
1	Slight: Some malformed or degraded cell under specimen
2	Mild: Zone limited to area under specimen
3	Moderate: Zone extending specimen size up to 1 cm
4	Severe: Zone extending farther than 1 cm

**Table 4 bioengineering-11-00720-t004:** Summary of molecular bonds associated with chitosan and tricalcium phosphate minerals.

Bond	Peak Type	Range (cm^−1^)	Ref.
O-H and N-H	Stretching vibrations due to intermolecular hydrogen bonding	3200–3500	[3,36,37,38,39]
C-H	Stretching vibrations in the CH_2_ and CH_3_	2870–2920	
C=O	Amide I: Stretching vibration in the amide group	1600–1650	
N-H and C-N	Amide II: Bending and stretching vibrations, respectively	1550–1590	
CH_3_	Symmetric bending vibrations	1370–1390	
C-O-C	Stretching vibrations in the glycosidic linkage	1150–1100	
C-O	Stretching vibrations in the pyranose ring	1030–1060	
υ_4_ ${PO}_{4}^{3-}$	Phosphate bending vibrations	560–600	[3,33,34,35]
υ_1_ ${PO}_{4}^{3-}$	Symmetric stretching vibrations	960–962	
υ_3_ ${PO}_{4}^{3-}$	Asymmetric stretching vibrations	1020–1100	

## Data Availability

The data used to support the findings of this study are available from the corresponding author upon reasonable request.

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
