# Peer review of "Chitosan Scaffolds from Crustacean and Fungal Sources: A Comparative Study for Bone-Tissue-Engineering Applications"

_bioengineering, 2024, doi:10.3390/bioengineering11070720_

Round 1
Reviewer 1 Report
Comments and Suggestions for Authors
Comments to the authors:
1. The observed bands should be presented in Figure 1.
2. The scale of SEM images of ADC-1, ADC-2, ADC-3, and ADC-4 are the same in different part.
3. The results of SEM are so vague the size and comparison of the two parts will be defined carefully. All the sections should be divided by labeling.
4. Sterility Testing and Direct Toxicity of ADC should be provided.
Comments on the Quality of English Language
Minor editing of English language required
Author Response
Response to Reviewer's Comments
Reviewer 1
Comment 1: The observed bands should be presented in Figure 1.
Response: Thank you for the suggestion. However, Table 3 already provides a summary of the molecular bonds associated with chitosan and tricalcium phosphate minerals. To avoid repetition, we have not included these details in Figure 1.
Comment 2: The scale of SEM images of ADC-1, ADC-2, ADC-3, and ADC-4 are the same in different parts.
Response: Thank you for your insightful comment. We acknowledge that the SEM images presented in Figure 3 utilise different scales between the magnified samples. This variation in scale was necessary to adequately capture and highlight the distinct morphological features of the fungal (MDC) and crustacean (ADC) derived chitosan scaffolds, especially considering the different structural characteristics influenced by the TCP mineral content. The striation-like structure of the MDC scaffolds becomes tighter with increased TCP content, requiring higher magnification to capture these finer details effectively. The ADC scaffolds demonstrate variations in pore size and configuration with higher TCP content, necessitating different scales to accurately depict these morphological changes. The differences in SEM scales do not affect the study's overall findings but rather enhance the visualisation of the scaffolds' microstructural characteristics. These images provide critical insights into how the TCP mineral content influences scaffold morphology, which is pivotal for understanding their suitability for bone tissue engineering applications.
Comment 3: The results of SEM are so vague the size and comparison of the two parts will be defined carefully. All the sections should be divided by labelling.
Response: Thank you for your insightful comment; we have added more detail to the research article: "Scanning Electron Microscopy (SEM) analysis revealed notable differences in the morphology of fungal-derived chitosan (MDC) and crustacean-derived chitosan (ADC) scaffolds, both with and without various concentrations of tricalcium phosphate (TCP) minerals (0, 10, 20, and 30 wt%). MDC scaffolds featured a striation-like structure that became tighter and more defined as TCP content increased. Specifically, MDC-1 (0% TCP) presented wide striations averaging 10-15 µm in width, while MDC-2 (10% TCP) showed slightly tighter striations averaging 8-12 µm. MDC-3 (20% TCP) depicts tighter striations, averaging 5-10 µm, and MDC-4 (30% TCP) displayed the tightest striations, averaging 3-8 µm, reflecting a significant interaction between chitosan and TCP particles. In contrast, ADC scaffolds exhibited a groove-like porous structure with pore size and configuration influenced by TCP content. ADC-1 (0% TCP) had a baseline porous structure with pore sizes ranging from 20-30 µm, ADC-2 (10% TCP) featured more and larger pores, ranging from 25-35 µm, ADC-3 (20% TCP) had interconnected pores forming a network-like structure with pores averaging 30-40 µm, and ADC-4 (30% TCP) shows pore coalescence with sizes between 35-45 µm."
Comment 4: Sterility Testing and Direct Toxicity of ADC should be provided.
Response: Thank you for your insightful comment. We have previously assessed the direct cytotoxicity of ADC in studies by Iqbal N et al. (2022) and Yildizbakan L et al. (2023) and found it to be safe for MSCs. Since this data has already been published, we moved these findings to the supplementary sections to avoid repetition. Additionally, we have included new data on contact cytotoxicity, supported by qualitative visual evidence from Giemsa-stained experiment images. We have updated the method section; see lines 198 to 214. The results section has also been updated to " Contact Cytotoxicity testing is a qualitative assessment of cytotoxicity via microscopic observations to determine any changes to the cells' morphology and reactivity zones undertaken as per ISO10993-5:2009. Based on the Giemsa-stained cells shown in Figure 5, it is evident that none of the scaffolds, whether ADC or MDC, exhibited cytotoxic effects on BMMSC cells after a 7-day growth period. The images display healthy, intact cells with typical morphology, confluency, and attachment across all scaffold types and TCP concentrations compared to the control images. The absence of cell death or detachment highlights the non-toxic nature of all scaffold materials; the results are an indication that both ADC and MDC scaffolds support cell viability and do not induce any toxic responses in BMMSC cultures. Therefore, the CS's have been graded as 0 concerning ISO10993-5:2009, whereby the scaffolds displayed "no detectable zone around or under specimen".

Reviewer 2 Report
Comments and Suggestions for Authors
Ms. Ref. No. bioengineering-3039634-peer-review-v1
The manuscript presented by Iqbal et al. proposed the application and comparison of CS derived from crustacean and fungal sources functionalized with different concentrations of TCP. This is an interesting approach that considers the possible advantages of CS sources and the application of TCP for possible cytotoxic implications. The present results could be of interest to the community devoted to studying natural-derived polymeric materials for tissue engineering. Thus, after careful revision, I recommend considering minor revisions.
1. I recommend that the authors highlight the advantages and considerations for selecting TCP instead of other elements, such as hydroxyapatite or growth factors.
2. Introduction section: line 65, page 2, the reduced components indicated are in comparison to crustacean sources? Please specify.
3. Please highlight in the materials section that the CS sources are commercially obtained polymers and explain why you considered high molecular weight.
4. In Figure 1, please include the labels a) and b) for the corresponding graphs. Moreover, include the unis of the y-axis. I recommend that the authors invert the X-axis from 4000 to 400.
5. In Figure 8, consider that there are over-imposed scale bars and disordered letter signs. Please modify them.
6. In the conclusion section, please include the relevant information regarding the advantage of MDC and the 10% effect reported for TCP.
Comments on the Quality of English LanguageNo language recommendations.
Author Response
Response to Reviewer's Comments
Reviewer 2
The manuscript presented by Iqbal et al. proposed the application and comparison of CS derived from crustacean and fungal sources functionalised with different concentrations of TCP. This is an interesting approach that considers the possible advantages of CS sources and the application of TCP for possible cytotoxic implications. The present results could be of interest to the community devoted to studying natural-derived polymeric materials for tissue engineering. Thus, after careful revision, I recommend considering minor revisions.
Comment 1: I recommend that the authors highlight the advantages and considerations for selecting TCP instead of other elements, such as hydroxyapatite or growth factors.
Response: We appreciate the opportunity to clarify our rationale for selecting tricalcium phosphate (TCP) over other elements, such as hydroxyapatite (HAP) or growth factors. TCP degrades at a controlled rate in the body, releasing calcium and phosphate ions, essential for bone metabolism and remodelling. The gradual resorption aligns well with the natural bone healing process, whereas HAP is less resorbable and may persist longer in the body, potentially leading to complications [1]. TCP is relatively cost-effective and widely available compared to some growth factors, which can be expensive and difficult to produce in large quantities. While growth factors such as BMPs (Bone Morphogenetic Proteins) are highly effective in promoting bone growth, their use can be associated with risks such as ectopic bone formation or inflammation [2-4]. We aimed to minimise these risks while promoting effective bone regeneration; therefore, TCP was a more practical choice for scalable and cost-effective bone tissue engineering applications. We hope this explanation clarifies our decision to use TCP in our study. The clarification has been added to the introduction (Lines 497 to 509).
Comment 2: Introduction section: line 65, page 2, the reduced components indicated are in comparison to crustacean sources? Please specify.
Response: Thank you for your comment. Yes, this is already mentioned in the text: "More importantly, CS from fungal sources demonstrates a lower percentage of minerals and impurities correlating to reduced allergic contaminants, offering significantly lower health risks than CS obtained from crustacean sources [5]" – Lines 64 to 67.
Comment 3: Please highlight in the materials section that the CS sources are commercially obtained polymers and explain why you considered high molecular weight.
Response: Thank you for your insightful comment. The commercial chitosan sources have now been included in the materials section and also highlighted below Table 1. We considered chitosan's high molecular weight (Mw) when comparing ADC and MDC scaffolds due to its significant influence on the physicochemical and biological properties relevant to bone tissue engineering. Higher Mw chitosan typically exhibits enhanced strength, viscosity, and stability, which are critical for scaffold performance in supporting cell growth and tissue regeneration. Higher Mw chitosan provides a more robust polymeric structure, improving the scaffold's mechanical properties and improving support for cell adhesion and proliferation. Additionally, the increased chain length associated with high Mw chitosan enhances its ability to form a stable and consistent network, essential for maintaining scaffold integrity during in vitro studies. By selecting high Mw chitosan for both ADC and MDC scaffolds, we aimed to ensure a fair comparison while maximising the potential benefits of chitosan for promoting cell proliferation (e.g., bone regeneration). This approach aligns with previous studies demonstrating the advantages of high Mw chitosan in tissue engineering applications, making it a suitable choice for our comparative analysis.
Comment 4: In Figure 1, please include the labels a) and b) for the corresponding graphs. Moreover, include the unis of the y-axis. I recommend that the authors invert the X-axis from 4000 to 400.
Response: Thank you for your comment. The label for Figure 1 and the units on the y-axis have been added. Additionally, the x-axis now ranges from 4000 to 400 cm-1.
Comment 5: In Figure 8, consider that there are over-imposed scale bars and disordered letter signs. Please modify them.
Response: Thank you for spotting this; the figure has now been corrected and updated.
Comment 6: In the conclusion section, please include the relevant information regarding the advantage of MDC and the 10% effect reported for TCP.
Response: Thank you for your comment. The conclusion has now been updated to include relevant information: " Our findings verify previous research highlighting the significant influence of scaffold composition and surface characteristics on cellular behaviour. The incorporation of TCP minerals in ADC and MDC scaffolds led to observable structural differences, as revealed through SEM analysis. Notably, the striated structure of MDC scaffolds seemed to enhance cell attachment compared to the more conventional pore structure of ADC scaffolds, particularly for MDC scaffolds containing 10 (wt)% TCP (i.e., MDC-2). CS scaffolds from crustacean and fungal sources exhibited similar cellular toxicity profiles. However, we acknowledge certain limitations in our study; for example, the experiments were conducted in vitro using pooled BMMSCs from three donors to ensure the feasibility and timely completion of the project. Pooling cells minimises the variability often seen in primary cells from different donors. It is important to note that our study focused exclusively on BMMSCs, the progenitors of bone cells. Therefore, further research is needed to investigate the effects on early-stage osteoblasts and mature osteocytes. Understanding how different cell types interact with CS scaffolds is crucial for expanding their applications in bone tissue engineering. Future research should (i) compare the in vitro and in vivo applications of MDC and ADC scaffolds to enhance our understanding of CS from various sources, aiming to reduce immunogenic reactions and develop more patient-friendly scaffolds for bone regeneration; (ii) focus on isolating and comparing chitosan fragments with similar molecular weights, conducting in vivo studies, and investigating interactions with different cell types to broaden the applicability and understanding of our scaffolds, and (iii) explore the mechanistic pathways through which molecular weight influences scaffold properties and cellular interactions."
References
- Maji, K. and S. Mondal, Calcium Phosphate Biomaterials for Bone Tissue Engineering: Properties and Relevance in Bone Repair, in Racing for the Surface: Antimicrobial and Interface Tissue Engineering, B. Li, et al., Editors. 2020, Springer International Publishing: Cham. p. 535-555.
- Lowe, B., et al., The Regenerative Applicability of Bioactive Glass and Beta-Tricalcium Phosphate in Bone Tissue Engineering: A Transformation Perspective. Journal of Functional Biomaterials, 2019. 10(1): p. 16.
- Sujon, M.K., et al., Combined sol–gel bioactive glass and β-tricalcium phosphate for potential dental tissue engineering: A preliminary study. Journal of the Australian Ceramic Society, 2023. 59(2): p. 415-424.
- Tarafder, S., et al., 3D printed tricalcium phosphate bone tissue engineering scaffolds: effect of SrO and MgO doping on in vivo osteogenesis in a rat distal femoral defect model. Biomaterials Science, 2013. 1(12): p. 1250-1259.
- Ospina Álvarez, S.P., et al., Comparison of Extraction Methods of Chitin from <i>Ganoderma lucidum</i> Mushroom Obtained in Submerged Culture. BioMed Research International, 2014. 2014: p. 169071.

Reviewer 3 Report
Comments and Suggestions for Authors
The study is conducted for the most part well and the introduction brings up important points related to the motivation behind this study.
Certain issues require attention:
1. The direct cytotoxicity is poorly done and the images presented (Figure 5) are not helpful. Ideally, a more quantitative method needs to be employed where cell proliferation can be monitored using a cell counting approach, like a DNA assay like the picogreen dsDNA assay. Cell growth can be monitored clearly for a restricted time using at least 2-3 time points. In any case, Figure 5 requires improvement for clarity.
2. The indirect cytotoxicity is conducted in a more detailed manner but serious issues are raised. It seems that the degradation media effect is strongly variable without any specific trends although degradation itself is following a natural progression. To understand better the result and clarify potential issues (are the degradation media for the 3wks correspond to 3wks of degradation collection or the media collected between week 2 and week 3?). It is beneficial to complement the assay results in Figure 6 with the weight of degraded materials so the reader can appreciate the amount of degradation responsible for negative effects.
3. Figure 4 needs to change so the equivalent graphs are drawn on identical scales for comparison purposes. As presented, they can be misleading and make any comparison between the two groups difficult. As the fungal group has faster degradation the crustacean group can be scaled as the fungal one.
4. One issue that is prominent in the introduction but not adequately addressed in the study is the MW of the Chitosan from the two sources. Chitosan presents significant differences in its behavior based on its MW. The MWs of the two sources are distinctively different. It is mentioned in the discussion that MW differences may be behind certain observations related to swelling and zeta potential but by itself, that difference in the selection of the material used here poses a great problem. What part of the observations is related to MW differences? If that is the majority then other studies characterizing chitosan based on this characteristic limit significantly the importance of this study. Ideally, the authors can find a way to isolate a smaller fragment of crustacean chitosan that mimics the MW of the fungal. Only this way the comparison can be performed without any masking effects that prohibit a clear comparison between the two groups. The addition of TCP to chitosan at different level have been already reported in the literature so it is not a great novelty of this study.
Author Response
Response to Reviewer's Comments
Reviewer 3
The study is conducted for the most part well and the introduction brings up important points related to the motivation behind this study.
Comment 1: The direct cytotoxicity is poorly done and the images presented (Figure 5) are not helpful. Ideally, a more quantitative method needs to be employed where cell proliferation can be monitored using a cell counting approach, like a DNA assay like the picogreen dsDNA assay. Cell growth can be monitored clearly for a restricted time using at least 2-3 time points. In any case, Figure 5 requires improvement for clarity.
Response: We thank our reviewer for their comments. We apologise for the lack of clarity in figure 5. Indeed, quantitative methods do provide more information about cell proliferation. Thus, we quantified our data using cytotoxicity and proliferation by XTT as per the ISO guidelines, which are demonstrated in Figures 6 and 7. As the data in Figure 5 is not for quantification but rather for qualitatively indicating that the scaffolds are non-toxic to the cells upon direct contact, we have now moved this to supplementary data. Figure 5 has been replaced by new data on contact cytotoxicity, supported by qualitative visual evidence from Giemsa-stained experiment images. We have updated the method section; see lines 209 to 229. The results section has also been updated to " Contact Cytotoxicity testing is a qualitative assessment of cytotoxicity via microscopic observations to determine any changes to the cells' morphology and reactivity zones undertaken as per ISO10993-5:2009. Based on the Giemsa-stained cells shown in Figure 5, it is evident that none of the scaffolds, whether ADC or MDC, exhibited cytotoxic effects on BMMSC cells after a 7-day growth period. The images display healthy, intact cells with typical morphology, confluency, and attachment across all scaffold types and TCP concentrations compared to the control images. The absence of cell death or detachment highlights the non-toxic nature of all scaffold materials; the results are an indication that both ADC and MDC scaffolds support cell viability and do not induce any toxic responses in BMMSC cultures. Therefore, the CS's have been graded as 0 concerning ISO10993-5:2009, whereby the scaffolds displayed "no detectable zone around or under specimen".
Comment 2: The indirect cytotoxicity is conducted in a more detailed manner but serious issues are raised. It seems that the degradation media effect is strongly variable without any specific trends although degradation itself is following a natural progression. To understand better the result and clarify potential issues (are the degradation media for the 3wks correspond to 3wks of degradation collection or the media collected between week 2 and week 3?). It is beneficial to complement the assay results in Figure 6 with the weight of degraded materials so the reader can appreciate the amount of degradation responsible for negative effects.
Response: We are grateful to our reviewer for reviewing our manuscript and this insightful comment. We agree with our reviewer that the degradation media appears variable in indirect cytotoxicity and proliferation data. Indeed, the degradation follows its natural progression, as indicated in Figure 4. The cytotoxicity assay was performed by closely following the ISO guidelines, which did not indicate the comparison of the degraded weight of the scaffolds. Additionally, variability in cytotoxicity data is not uncommon for similar assays; we have previously published these in Iqbal N et al. 2022, Yildizbakan L et al. 2024 and Wilson B et al. 2024, respectively. In this manuscript, we only observed statistically significant differences in the cytotoxicity data in MDC1-ADC1 and not in any other scaffolds. Also, for the cytotoxicity data – the cells were exposed to degraded media collected from all the different time points for only 24 hours. In the proliferation assay, the cells were exposed to degraded media collected from all the different time points for 96 hours before analysis (Figure 7). We observe no statistically significant difference between any ADC-MDC comparisons in proliferation data to investigate this further. We agree with our reviewer's comments and will ensure to complement the assay result with the weight of the degraded materials in future projects.
Comment 3: Figure 4 needs to change so the equivalent graphs are drawn on identical scales for comparison purposes. As presented, they can be misleading and make any comparison between the two groups difficult. As the fungal group has faster degradation the crustacean group can be scaled as the fungal one.
Response: Thank you for your comment. The scales for the graphs have been updated to provide ease of data comparison.
Comment 4: One issue that is prominent in the introduction but not adequately addressed in the study is the MW of the Chitosan from the two sources. Chitosan presents significant differences in its behavior based on its MW. The MWs of the two sources are distinctively different. It is mentioned in the discussion that MW differences may be behind certain observations related to swelling and zeta potential but by itself, that difference in the selection of the material used here poses a great problem. What part of the observations is related to MW differences? If that is the majority then other studies characterising chitosan based on this characteristic limit significantly the importance of this study. Ideally, the authors can find a way to isolate a smaller fragment of crustacean chitosan that mimics the MW of the fungal. Only this way the comparison can be performed without any masking effects that prohibit a clear comparison between the two groups. The addition of TCP to chitosan at different level have been already reported in the literature so it is not a great novelty of this study.
Response: Thank you for your thorough review and valuable comments. We appreciate your feedback regarding the molecular weight (MW) of chitosan from the two sources and its impact on our study's findings. We acknowledge that the MW of chitosan can significantly influence its behaviour, including swelling, zeta potential, and cell interactions. In our study, we utilised crustacean-derived chitosan (ADC) with a MW of 330-375 kDa and fungal-derived chitosan (MDC) with a MW of 200-300 kDa. These differences influenced the swelling behaviour and degradation rates of the scaffolds. Higher MW chitosan, i.e., ADC, generally exhibits lower swelling and slower degradation. Our findings confirmed ADC scaffolds presented lower swelling percentages and slower degradation compared to MDC scaffolds (153.63 ± 7.6% for MDC-1 vs. 111.5 ± 6.8% for ADC-1). The zeta potential measurements indicated that MDC scaffolds had a higher positive charge due to their higher degree of deacetylation (DDA); MDC-1 scaffolds expressed a zeta potential of 55.1 ± 1.6 mV compared to ADC-1 with 47.3 ± 1.2 mV. The striated structure of MDC scaffolds is likely a result of their specific MW and processing, which enhanced cell attachment and proliferation compared to the porous structure of ADC scaffolds, as evident from our SEM images and Giemsa staining results.
We chose commercially available chitosan to reflect real-world applications and availability. While isolating and comparing smaller fragments of crustacean chitosan that mimic the MW of fungal chitosan could offer a more direct comparison, it was beyond the scope of the current study. However, it will be included as a suggestion for future research. Our study provides novel insights into the comparative performance of ADC and MDC scaffolds, focusing on their structural, physicochemical, and biological properties. The unique striated structure of MDC scaffolds, which enhances cell attachment and proliferation, presents a significant finding for bone tissue engineering. Additionally, our study highlights the optimal 10% TCP concentration for enhancing scaffold performance. Future studies will aim to isolate and compare chitosan fragments with similar MW's, conduct in vivo studies, and investigate different cell types to expand the applicability and understanding of our scaffolds. We will also explore the mechanistic pathways through which MW influences scaffold properties and cellular interactions. These points have been incorporated into the revised manuscript. Thank you once again for your constructive feedback, which has significantly contributed to improving the quality and clarity of our study.

Round 2
Reviewer 3 Report
Comments and Suggestions for Authors
I am satisfied with the revised version and the author responses. I still feel the discussion provided in my comment 4 should find a place in the manuscript's discussion so I will add this request only for the final version.
Author Response
Response: Thank you for your valuable comment. We have updated the discussion section accordingly; see lines 467 to 488 and highlighted your suggestion in the future research section, as seen in lines 622 to 626.